# Modeling of core-shell magneto-electric nanoparticles for biomedical applications: Effect of composition, dimension, and magnetic field features on magnetoelectric response

Serena Fiocchi (ORCID) *[�], Emma Chiaramello[�], Alessandra Marrella, Giulia Suarato, Marta Bonato, Marta Parazzini (ORCID), Paolo Ravazzani

Institute of Electronics, Information Engineering and Telecommunications (IEIIT), National Research Council of Italy (CNR), Turin, Italy

euro These authors contributed equally to this work.
* serena.fiocchi@ieiit.cnr.it

## Abstract

The recent development of core-shell nanoparticles which combine strain coupled magnetostrictive and piezoelectric phases, has attracted a lot of attention due to their ability to yield strong magnetoelectric effect even at room temperature, thus making them a promising tool to enable biomedical applications. To fully exploit their potentialities and to adapt their use to *in vivo* applications, this study analyzes, through a numerical approach, their magnetoelectric behavior, shortly quantified by the magnetoelectric coupling coefficient ($\alpha_{ME}$), thus providing an important milestone for the characterization of the magnetoelectric effect at the nanoscale. In view of recent evidence showing that $\alpha_{ME}$ is strongly affected by both the applied magnetic field DC bias and AC frequency, this study implements a nonlinear model, based on magnetic hysteresis, to describe the responses of two different core-shell nanoparticles to various magnetic field excitation stimuli. The proposed model is also used to evaluate to which extent realistic variables such as core diameter and shell thickness affect the electric output. Results prove that $\alpha_{ME}$ of 80 nm cobalt ferrite-barium titanate (CFO-BTO) nanoparticles with a 60:40 ratio is equal to about 0.28 V/cm·Oe corresponding to electric fields up to about 1000 V/cm when a strong DC bias is applied. However, the same electric output can be obtained even in absence of DC field with very low AC fields, by exploiting the hysteretic characteristics of the same composites. The analysis of core and shell dimension is as such to indicate that, to maximize $\alpha_{ME}$, larger core diameter and thinner shell nanoparticles should be preferred. These results, taken together, suggest that it is possible to tune magnetoelectric nanoparticles electric responses by controlling their composition and their size, thus opening the opportunity to adapt their structure on the specific application to pursue.

**Data Availability Statement:** All data are present within the paper and/or Supporting Information files.

**Funding:** The authors received no specific funding for this work.

**Competing interests:** The authors have declared that no competing interests exist.

# 1. Introduction

Discoveries of micro and nanoparticle (NP) properties and their formulations have dominated the biomedical field in the last few decades. This has resulted in tremendous efforts by researchers to keep designing and developing new smart materials which can be studied for their unique properties and synthesized into nanomaterials with exceptional functional abilities.

For example, the investigation and applications of nanoscale liquids, liquid interfaces, and fluids, briefly referred to as nanofluidic [1], has provided revolutionary solutions for handling extremely small volumes of liquids, thus representing a powerful tool for the study of molecular behaviors, intermolecular interactions, and molecule-environment interactions in solution [2–7]. Similarly, magnetic nanoparticles have drawn exceptional attention from different technological applications due to their high chemical stability with enhanced surface area, biocompatibility, tunable magnetic moment and functionalization with polymers and other materials [8,9].

As a natural evolution of magnetic materials and core-shell technologies, a few years ago, researchers have developed a next-generation core-shell nanoparticles with unique properties given by the so called magnetoelectric (ME) effect [10], which originates at the interface between the magnetostrictive ferrite core and the piezoelectric shell and results in a strain mediated coupling between the applied magnetic field and the generated electric field.

In other words, magneto-electric nanoparticles (MENPs) can wirelessly convert externally applied magnetic field into electric field at the nanoscale and this makes them an extremely innovative and valuable candidate for manipulating electric fields at cellular and subcellular levels. Their use as wireless nanoelectrodes could indeed enable a wide range of applications in which their energy efficiency and the lack of invasiveness represent tremendous advantages over currently used approaches. This is the special case of brain stimulation. It is noteworthy that electric-field triggered stimulation is the basis for many modern stimulation techniques such as invasive direct-contact deep-brain stimulation (DBS) and low-efficacy transcranial magnetic stimulation (TMS). However, these approaches are severely limited in their potentials. The DBS entails direct physical contacts with the brain tissues and is therefore constrained to a finite number of implants. TMS is connected only indirectly with neural networks, and therefore, has very low efficiency and poor spatial resolution. In contrast, wireless ME-based stimulation can be performed locally, and therefore, can be completely non-invasive (or only minimally invasive) while achieving extraordinarily high efficacy.

Other possible biomedical applications that can benefit from such innovative and unique nanocomposites include neural recording, nano electroporation, cancer treatment, drug delivery, tissue engineering and imaging (for a complete survey of potential uses in biomedical field see the review studies [11–13]).

The most common composite used in these early applications of MENPs is made of a combination of cobalt ferrite (CFO) and barium titanate (BTO) due to its good biocompatibility and high magnetoelectric coupling coefficient even at room temperature [14,15], but other compounds such as iron oxide (FO)/BTO and Nickel/BTO are emerging as possible valuable candidates to broad the use cases [16].

Characterization of such low scale ME nanocomposites is often not possible due to the difficulty in the direct ME measurements, the necessity of measuring a wide range of magnetic fields and the need of studying high particle concentration effects [17,18].

In this context, Finite element modeling (FEM) has been proven as a relevant tool to characterize different ranges of magnetoelectric geometric arrangements and structures [18,19] and can provide a better understanding of which MENPs properties mostly influence the magnetoelectric coupling.

In case of direct ME effect (i.e., when the electric field is elicited by an external magnetic field), MENPs response is physically characterized by the ME coefficient ($\alpha_{ME}$), which quantifies the coupling efficiency between the change in the external applied magnetic field, both at DC and AC frequency, and the generated electric field. Commonly reported experimental $\alpha_{ME}$ values range between 0.001 and 0.1 V/cm/Oe [12,20], but recent studies claimed $\alpha_{ME}$ values even above 1 V/cm/Oe [21,22]. Reasons of this variability are either due to factors related to the fabrication process (e.g. mechanical mismatch between the core and the shell, dependence of the strain transfer at the interface on the sintering temperature, current leakage due to the low resistivity in the ferrite phase) and due to the physical characteristics of the two phases in different experimental conditions (volume material composition, different response with varying magnetic field frequencies, hysteretic behavior) [23]. If the former can be mainly controlled and mitigated through the advancement in nanoparticles engineering, the latter can be systematically and effectively characterized through computational methods. To the best of the authors knowledge, a proper in silico model that allows to fully assess different core-shell nanoparticles performance under variable settings is still lacking in the scientific literature and this could preclude the determination of the conditions whereby MENPs can be used. This paper then aims to fill this gap by modelling MENPs behavior at the nanoscale thorough a FE method and lays the theoretical framework for interpreting experimental results and, in perspective, tuning MENPs properties according to their intended use in biomedical field.

In particular this study relies on the following peculiar experimental findings on, even different, ME composites for tuning the ME coupling coefficient: a) the hysteresis loop of ME composites can be leveraged to induce the ME effect even when the DC magnetic field, which is used to magnetize them, is removed ("self-biased" ME composites) [24–26]. As a consequence, different core materials, with different hysteretic properties, differently affect the ME coefficient; b) core volume and surface shell thickness, as well as their relative dimensions, [14,18,19,27,28] play a crucial role in all the phases of magnetoelectric process (i.e. core magnetization and magnetostriction, strain mediated coupling at the core-shell interface, piezoelectric response); c) even low amplitude applied AC magnetic fields at frequencies far from the acoustic resonance (> 5 GHz), as in the near-DC frequency where most of biomedical applications operate, yield a strong ME effect.

This paper proposes a theoretical frame for studying and understanding the ME response of MENPs when these three experimental conditions are varied. Specifically, the three different analyses conducted in this study can be summarized as follows:

a. The behavior of two different core materials (i.e., CFO and FO) when subjected to varied DC magnetic field bias was investigated in terms of core magnetization, which is the start engine of the magnetostrictive process. This first analysis allows to assess how the selection of different core materials could drive the choice of the excitation magnetic field settings and could be particularly useful in future *in vivo* applications, to properly fit experimental studies constraints (such as power and safety limits).

b. Then, the mechanical quantities related to the core magnetostriction, transferred to the piezoelectric shell, and the mechanically induced electric response of the shell itself were calculated under a high strength DC magnetic field (i.e., when the core is fully magnetized and has reached magnetic saturation) and for variable core-shell sizes. This allowed to precisely assess the influence of MENPs material composition and size on the resulting magnetoelectric coefficient.

c. As a last step, according to the recent use of CFO-BTO MENPs in biomedical applications under low amplitude AC magnetic field excitation (see e.g. [29–34]), a possible explanation

of the colossal magnetoelectric response which implies the positive experimental outcomes found even at those weak excitation fields was provided.

The results of these analyses lay the basis for a better understanding of the impact of core-shell MENPs properties and provide an analytical framework to assess the MENPs behavior in biomedical research, with an eye to their possible application to brain stimulation.

## 2. Materials and methods

A schematic summary of the methodology applied in this study is shown in Fig 1 and detailed in the following paragraphs.

### 2.1 Nanoparticles modeling

A finite element method-based simulation tool, COMSOL Multiphysics 5.6 [35], was used to characterize the magnetoelectric behavior of two core-shell nanoparticles that have been recently synthetized for *in vivo* and *in vitro* experiments and have been proven to be biocompatible.

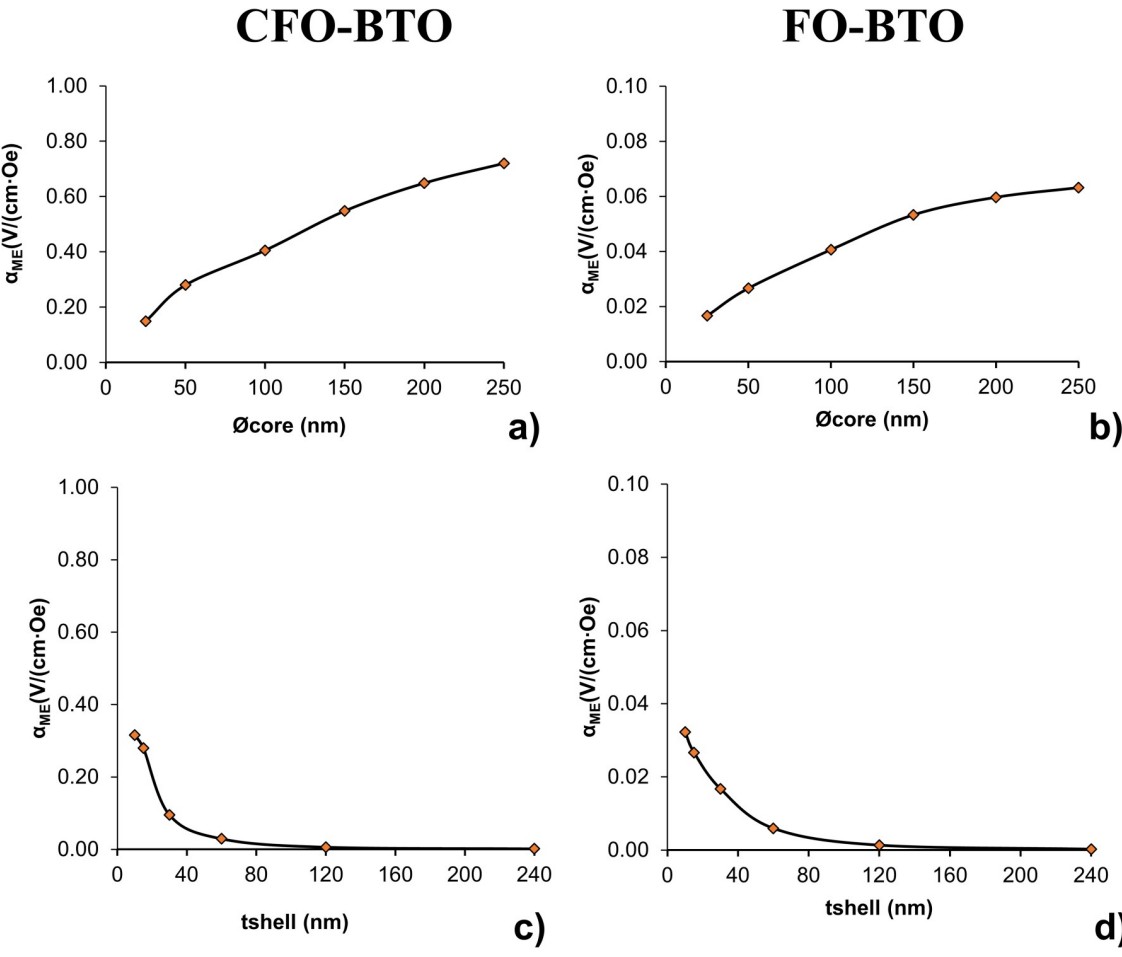

**Fig 1. MENP computational modeling.** Schematic representation of: a) the geometrical parameters of a generic core-shell MENP; b) the simulation settings in the three different analyses performed; c) the computational study workflow.

**Table 1. MENPs core properties.**

| | Symbol | CFO | FO | Reference |
|---|---|---|---|---|
| Core Diameter (nm) | $\varnothing_{core}$ | 50, range: 25–250 | 50, range: 25–250 | [30,36] |
| Magnetic saturation (A/m) | $M_s$ | $2.47*10^5$ | $2.88*10^5$ | [36,37] |
| Magnetization reversibility | $c$ | 0.4 | 0.95 | [38,39] |
| Domain wall density (A/m) | $a$ | $1*10^5$ | $1*10^5$ | [38,39] |
| Interdomain coupling | $\alpha$ | 1.4 | 0.5 | [38,39] |
| Pinning loss (A/m) | $k$ | $2*10^5$ | $5*10^5$ | [38,39] |
| Saturation magnetostriction (ppm) | $\lambda s$ | -200 | -20 | [36,40] |
| Density (kg/m³) | $\rho$ | 5200 | 5170 | [41], COMSOL library |
| Initial magnetic susceptibility | $X_0$ | 3 | 60 | [36],COMSOL library |
| Electrical conductivity (S/m) | $\sigma$ | $5.2*10^6$ | 0.13 | COMSOL library |
| Poisson ratio | $\nu$ | 0.48 | 0.3 | [40], COMSOL library |
| Young's modulus (GPa) | $E$ | 230 | 2300 | [40,42] |

They are both coated by a piezoelectric barium titanate shell (BaTiO$_3$, in the following "BTO") and differ in the magnetostrictive core materials: i) Cobalt ferrite (CoFe$_2$O$_4$, in the following "CFO"); ii) Magnetite (Fe$_3$O$_4$, in the following "FO"), both considered among the most promising candidates for medical applications due to their unique physical and mechanical properties and great chemical and thermal stability.

They were modelled by an axisymmetric bi-dimensional (2-D) model of spheres (see Fig 1A), and since their size can be controlled in a range from few tens of nanometers up to more than 200 nm [13], here the electric output (i.e. magnetoelectric coefficient $\alpha_{ME}$) was calculated by alternatively varying the dimension of the core (i.e. core diameter $\varnothing_{core}$) and the shell (i.e. shell thickness $t_{shell}$), starting from the estimated size of recently synthetized MENPs [36].

Properties of the two different core materials and the BTO shell are reported in Tables 1 and 2, respectively. Most of them are taken from recently synthesized nanoparticles for *in vitro* and *in vivo* experimental studies for brain stimulation and drug delivery [30,36,37].

## 2.2 COMSOL Multiphysics equations

In the following the equations governing the magnetoelectric behavior of the nanoparticle as solved by COMSOL Multiphysics [35] are summarized. In synthesis, the Magnetic Fields, Solid Mechanics, and Electrostatics modules were used, along with Magnetostriction and Piezoelectric Effect Multiphysics couplings [43] (see Fig 1B).

**Table 2. Shell properties (from COMSOL library [35]).**

| | Symbol | BTO |
|---|---|---|
| Shell Thickness (nm) | $t_{shell}$ | 15, range: 10–120 |
| Density (kg/m³) | $\rho$ | 5700 |
| Relative permittivity | $\{\varepsilon_{r11}; \varepsilon_{r22}; \varepsilon_{r33}\}$ | {115.1;115.1; 1251.3} |
| Electrical conductivity (S/m) | $\sigma$ | 178.5 |
| Elasticity matrix, Voigt notation (GPa) | {cE11; cE12; cE22; cE13; cE23; cE33; cE14; cE24; cE34; cE44; cE15; cE25; cE35; cE45; cE55; cE16; cE26; cE36; cE46; cE56; cE66} | {150.4;65.6;150.4;65.; 65.9; 145.5; 0; 0;0;43.9;0; 0; 0;43.9; 0; 0; 0; 0; 0; 42.4} |
| Piezoelectric coupling matrix, Voigt notation (C/m²) | {eES11; eES21; eES31; eES12; eES22; eES32; eES12; eES23; eES33; eES14; eES24; eES34; eES15; eES25; eES35; eES16; eES26; eES36} | {0; 0; -4.32; 0; 0; -4.32; 0; 0; 17.4; 0; 11.4; 0; 11.4; 0; 0; 0; 0; 0} |

**2.2.1 Magnetic field physics.** Magnetic field modeling was based on Ampere's Law equations [44]:

$$\nabla \times \mathbf{H} = \boldsymbol{J} \tag{1}$$

$$\mathbf{B} = \nabla \times \mathbf{A} \tag{2}$$

$$\mathbf{J} = \sigma \boldsymbol{E} \tag{3}$$

and constitutive equation:

$$\mathbf{B} = \mu_0(\boldsymbol{H} + \boldsymbol{M}) \tag{4}$$

Where $H$ is the magnetic field, $B$ the magnetic flux density, $A$ the magnetic vector potential, $J$ the current density, $E$ is the electric field and $M$ is the magnetization.

When the magnetic core of the magnetostrictive material is subjected to an external field $\mathbf{H}$, it becomes magnetized according to the hysteresis loop.

One of the most used models to describe the magnetization dependence on the applied magnetic field M(H) of magnetic materials is the Jiles–Atherton (J–A) model [45].

According to the J-A theory, the magnetization $M$ can be decomposed into an irreversible magnetization, $M_{irr}$, due to discontinuities in the material structure, and a reversible magnetization, $M_{rev}$, due to the elastic bending of magnetic domain wall:

$$M = M_{rev} + M_{irr} \tag{5}$$

The irreversible component of the magnetization is given by the following differential equation:

$$\frac{dM_{irr}}{dt} = g\left(\frac{M_{rev}}{ck} \cdot \frac{dH_e}{dt}\right)\frac{M_{rev}}{|M_{rev}|} \tag{6}$$

Where $c$ is a measure of the magnetization reversibility, $k$ is the pinning loss which is proportional to the energy dissipation when a domain wall passes a pinning and g = 1 if $dH/dt>0$ and g = -1 if $dH/dt<0$.

Equations governing the J-A model [45] are as follows:

$$M_{rev} = c(M_{an} - M_{irr}) \tag{7}$$

where, $M_{an}$ is the anhysteretic magnetization, given by the Langevin's function:

$$M_{an} = M_s\left(\coth\frac{|H_e|}{a} - \frac{a}{|H_e|}\right)\frac{H_e}{|H_e|} \tag{8}$$

where $a$ is proportional to the magnetic domain density, $M_s$ is the saturation magnetization and $H_e$ is the effective magnetic field inside the magnetic core, which is given by:

$$H_e = H + \alpha M + \frac{3\lambda_s}{\mu_0 M_s^2}SM \tag{9}$$

where $\alpha$ is a measure of the interdomain coupling due to the hysteresis, whereas the last term represents the mechanical stress contribution to the material magnetization, which is called "Villari effect". This term is function of the magnetostriction coefficient $\lambda_s$, the saturation magnetization $M_s$, and the deviatoric stress tensor $S$, which is computed in the solid mechanics module by solving the equation shown in the next paragraph (Eqs (10) and (11)).

**2.2.2 Solid mechanics physics.**   Solid mechanics module [46] solves:

$$0 = \nabla \cdot \mathbf{S} + \boldsymbol{F}_v \tag{10}$$

and strain-displacement equation

$$\boldsymbol{\varepsilon} = \frac{1}{2}\left[(\nabla \mathbf{u})^T + \nabla \mathbf{u}\right] \tag{11}$$

where $\boldsymbol{F}_v$ is the volume deformation tensor, $\boldsymbol{u}$ is the solid displacement vector and $\boldsymbol{\varepsilon}$ is the strain tensor. Stress-strain constitutive equations vary by domain as described below (Eqs 12 and 14).

Stress and strain in the magnetostrictive material are related by the Hooke's law as:

$$\boldsymbol{S} = S_0 + \boldsymbol{c}_H : [\boldsymbol{\varepsilon} - \boldsymbol{\varepsilon}_{me}] \tag{12}$$

where $\boldsymbol{\varepsilon}_{\mathbf{me}}$ represent the magnetostrictive strain and for isotropic materials is modelled as the following quadratic isotropic form of the magnetization field:

$$\boldsymbol{\varepsilon}_{me} = \frac{3\lambda_s}{\mu_0 M_s^2} dev(\boldsymbol{M} \otimes \boldsymbol{M}) \tag{13}$$

and the stiffness matrix $c_H$ can be represented in terms of two parameters using the Young's modulus $E$ and Poisson's ratio $v$.

The piezoelectric stress is modelled by:

$$\boldsymbol{S} = S_0 + \boldsymbol{C} : \boldsymbol{\varepsilon} + \boldsymbol{E} \cdot \boldsymbol{e} \tag{14}$$

where $\mathbf{C}$ is the elastic right Cauchy deformation tensor, $\boldsymbol{E}$ is the electric field and $\boldsymbol{e}$ is the piezo-electric Voigt coupling matrix representing the stress tensor.

These quantities are linked to the electrostatic module by using piezoelectric coupling and solving Gauss' law as described in the next paragraph.

**2.2.3 Electrostatic physics.**   Gauss' Law [44] was solved in the electrostatic module:

$$\nabla \cdot \boldsymbol{D} = \rho_v \tag{15}$$

$$\mathbf{E} = -\nabla V \tag{16}$$

where $\boldsymbol{D}$ is the electric flux density and $\rho_v$ is the volume charge density.

In the piezoelectric materials the electric flux density is linked to the solid mechanics equation [47] according to:

$$\mathrm{D} = \varepsilon_0 \mathbf{E} + \varepsilon_0 \chi \mathbf{E} + \boldsymbol{e} : \boldsymbol{\varepsilon} \tag{17}$$

where $\varepsilon_0$ is the vacuum electric permittivity and $\chi$ is the relative electrical susceptibility.

Finally, the electric output of each NP was quantified through the magnetoelectric coefficient, which is the ratio between the maximum $\boldsymbol{E}$ field intensity at the outer border of the NP and the change in external magnetic field $\boldsymbol{H}$:

$$\alpha_{ME} = \Delta E/\Delta H. \tag{18}$$

The workflow shown in Fig 1C summarizes the physics solved in the study and the main quantities used in the different study steps performed to assess the MENPs electric output.

### 2.3 Simulations and magnetoelectric coefficient analysis

As discussed in the introduction and summarized in Fig 1B, three different analyses were performed using the whole or part of the model detailed above, with the ultimate scope of characterizing MENPs behavior under variable conditions.

In particular, a) first simulations were conducted to assess the behavior of the two different core materials in the magnetization process and then to characterize their hysteretic behavior. To do that, the magnetic field **H** was directed along the z-axis of the two 50 nm spherical core nanoparticles and its amplitude was linearly varied in the range ±12 kOe to determine the M (H) loop.

To assess the influence of variable core and shell size on the magnetoelectric coefficient of the two core-shell nanoparticles (b), stationary studies with a magnetic field above magnetic saturation (H = Ms) were conducted, according to realistic experimental conditions where a DC bias generated by permanent magnet or electromagnets is used to elicit the electric output [30,31,33,48]. Core size diameter $\emptyset_{core}$ was varied in the range 25–250 nm, considering a fixed shell thickness $t_{shell}$ equal to 15 nm, corresponding to a core volume fraction ranging between 45% and 90%. Similarly, shell thickness of 50 nm core size MENPs was varied between 10 and 120 nm, corresponding to a core volume fraction ranging between 70% and 9%.

As a last step, (c) a time-dependent analysis was performed to characterize the behavior of MENPs, in terms of magnetoelectric coefficient, when an AC magnetic field directed along the z-axis at low frequency and low amplitude (f = 50 Hz, 100 Oe) is applied to a pre-magnetized nanoparticle through a previous high amplitude DC bias excitation, according to their recent use in biomedical applications.

## 3. Results

### 3.1 Core magnetization behavior

As a first step for MENPs characterization, Fig 2 shows the magnetization curves for the two core (diameter $\emptyset_{core}$ = 50 nm) materials, under varying magnetic field intensities, according to the Jiles–Atherton (J–A) model.

The cobalt ferrite (CFO) nanoparticle, as expected, shows a clear hysteretic behavior, with a coercivity field of about $H_c$ = 1.7 kOe and a magnetic remanence $M_r$ = 22 emu/g, as reported

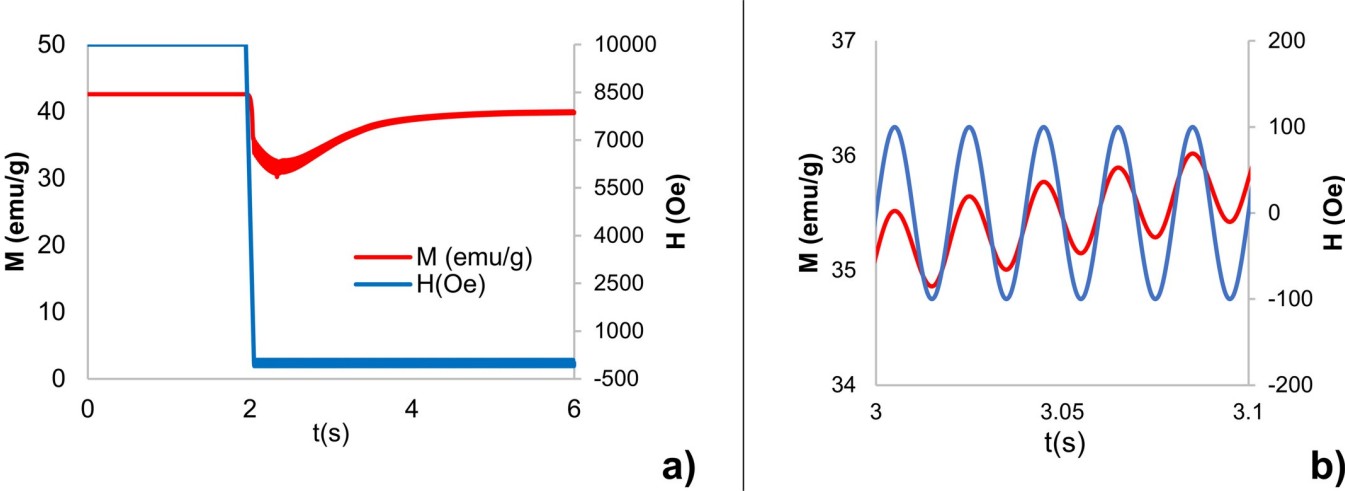

**Fig 2.** MENPs cores magnetization behavior. DC magnetization loops of a) CFO and b) FO core 50 nm nanoparticles.

in a recent experimental study performed on nanoparticles of the same size [36]. Moreover, it could be noted that the CFO magnetic core does not reach its saturation magnetization ($M_s$ = 47.5 emu/g) even at applied magnetic fields higher than 10 kOe.

In contrast, the FO nanoparticle has a superparamagnetic behavior, i.e. its M-H loop shows no hysteresis as reported in experimental studies of NPs of similar dimensions [37,49], and reaches magnetization near saturation (Ms = 55.7 emu/g) at magnetic field above few kOe.

### 3.2 DC charachterization of core-shell MENPs

Fig 3 shows the magnetization M (emu/g), the strain $\varepsilon$ (ppm), the electric field module E (V/m) and the potenial V (mV) on the surface of both CFO-BTO (left) and FO-BTO (right) core-shell nanoparticles ($\emptyset_{core}$ = 50 nm and $t_{shell}$ = 15 nm) when a DC magnetic field high enough to reach magnetic saturation in the core ($H_{DC}$ = $M_s$) is applied to elicit the magnetoelectric behavior.

As shown in the previous Fig 2 and Table 1, the two nanoparticles present similar saturation magnetization (i.e., $M_s$ = 47.5 and 55.7 emu/g for CFO and FO, respectively), but have saturation magnetostriction coefficients $\lambda_s$ which differ one order of magnitude (see Table 1). This translates in a larger strain transferred through the core-shell interface to the BTO shell when the core is made of CFO, as shown in Fig 3B. Similarly, the electric field and the resulting potential difference between the z-poles A and B (see Fig 1A) is about one order of magnitude greater for CFO-BTO than for FO-BTO, when all the other MENPs properties and stimulation conditions are kept constant.

As described above, the effect of core and shell size on magnetoelectric coefficient was quantified by varying core and shell thickness starting from a "standard" 50 nm core diameter —15 nm shell thickness nanoparticle. Results of these variations on the ME coefficient ($\alpha_{ME}$) for the two MENPs are shown in Fig 4.

ME coefficients show a similar behavior in response to variable core and shell thickness. Specifically, increasing core sizes translate in an increase of magnetoelectric coefficient which ranges from about 0.15 V/cm·Oe to 0.28 V/cm Oe, and to 0.72 V/cm·Oe, for 25 nm, 50 nm and 250 nm CFO core diameters, respectively, and from about 0.019 V/cm·Oe to 0.032 V/cm Oe, and to 0.074 V/cm·Oe, for 25 nm, 50 nm and 250 nm FO core diameters, respectively. Interestingly, there is a decreasing incremental ratio (i.e. decreasing curves slope) when increasing core size: as an example, ratios between the CFO-BTO ME coefficient for core diameters equal to 25 nm, 100 nm, 150 nm, 200 nm, 250 nm and the "standard" core size 50 nm are equal to 0.5, 1.4, 2.0, 2.3 and 2.6, respectively for CFO-BTO nanoparticle and 0.6, 1.5, 2.0, 2.2 and 2.4, respectively, for FO-BTO nanoparticle.

An opposite trend, but still similar for the two core materials MENPs, was found when increasing shell thickness: ME coefficient decreases with increasing shell thickness from about 0.32 V/cm·Oe to 0.095 V/cm Oe, and to 0.006 V/cm·Oe, for 10 nm, 30 nm and 120 nm CFO-BTO shell thickness, respectively, and from about 0.038 V/cm·Oe to 0.019 V/cm Oe, and to 0.002 V/cm·Oe, for 10 nm, 30 nm. and 120 nm FO-BTO shell thickness, respectively. In this case the incremental ratio between the CFO-BTO ME coefficient for shell thickness equal to 10 nm, 30 nm, 60 nm, 120 nm, 240 nm and the "standard" 15 nm shell thickness are equal to 1.13, 0.34, 0.1, 0.02 and 0.01, respectively for CFO-BTO nanoparticle and 1.21, 0.63, 0.22, 0.05 and 0.01, respectively, for FO-BTO nanoparticle.

### 3.3 AC charachterization of core-shell MENPs

Fig 5A shows the core magnetization M(emu/g) (red line), when the "standard" (i.e. $\emptyset_{core}$ = 50 nm and $t_{shell}$ = 15 nm) CFO-BTO MENP is subjected to a time-variant external magnetic field

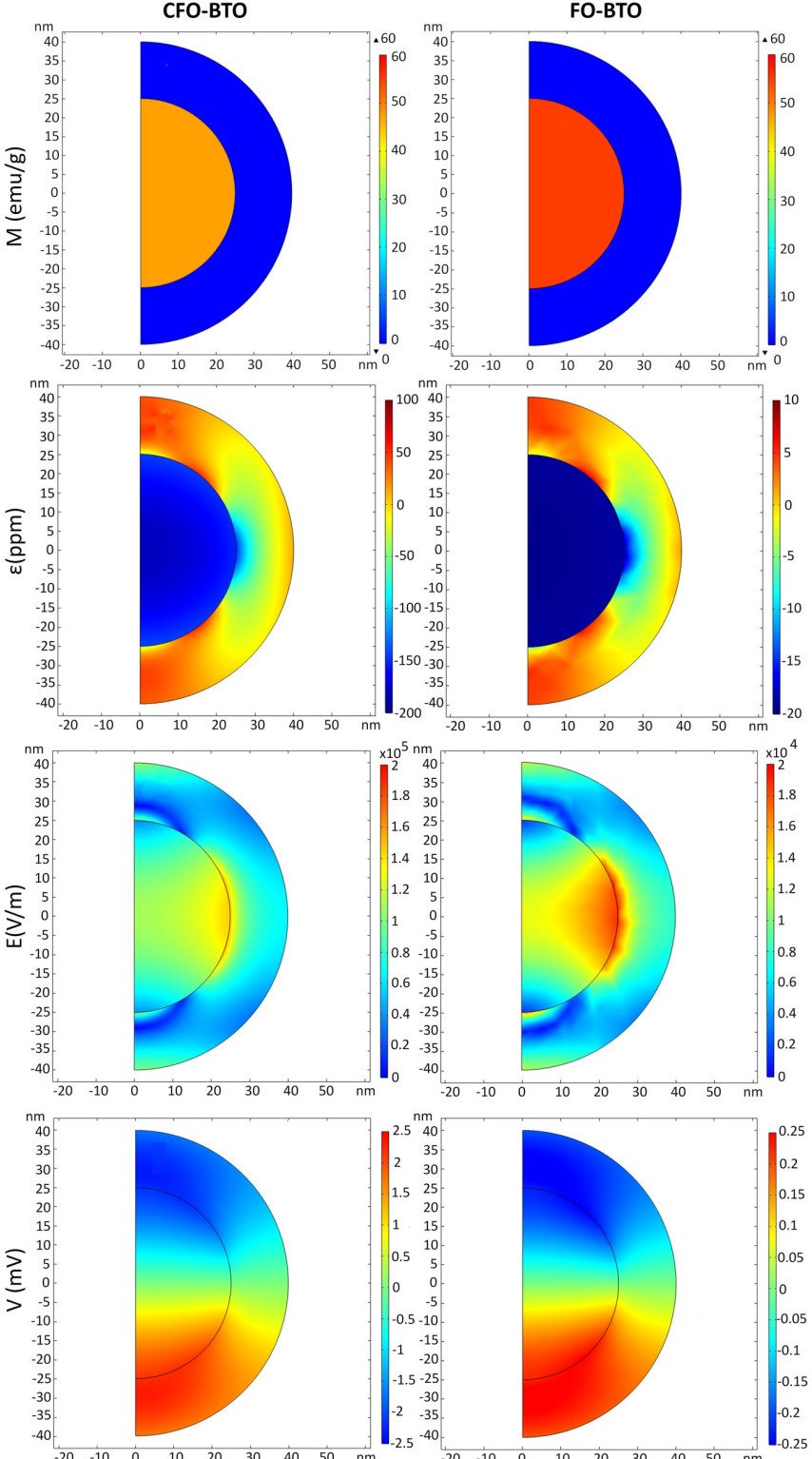

**Fig 3. Magnetoelectric effect elicited by DC magnetic field stimulation.** Distribution of (from top to bottom) surface: M (emu/g), strain $\varepsilon$ (ppm), electric field E module (V/m), and electric potential V (mV) in 2D axisymmetric (left) CFO-BTO and (right) FO-BTO core shell nanoparticles ($\varnothing_{core}$ = 50 nm and $t_{shell}$ = 15 nm) when a high amplitude (H = $M_s$) magnetic field is used as stimulation.

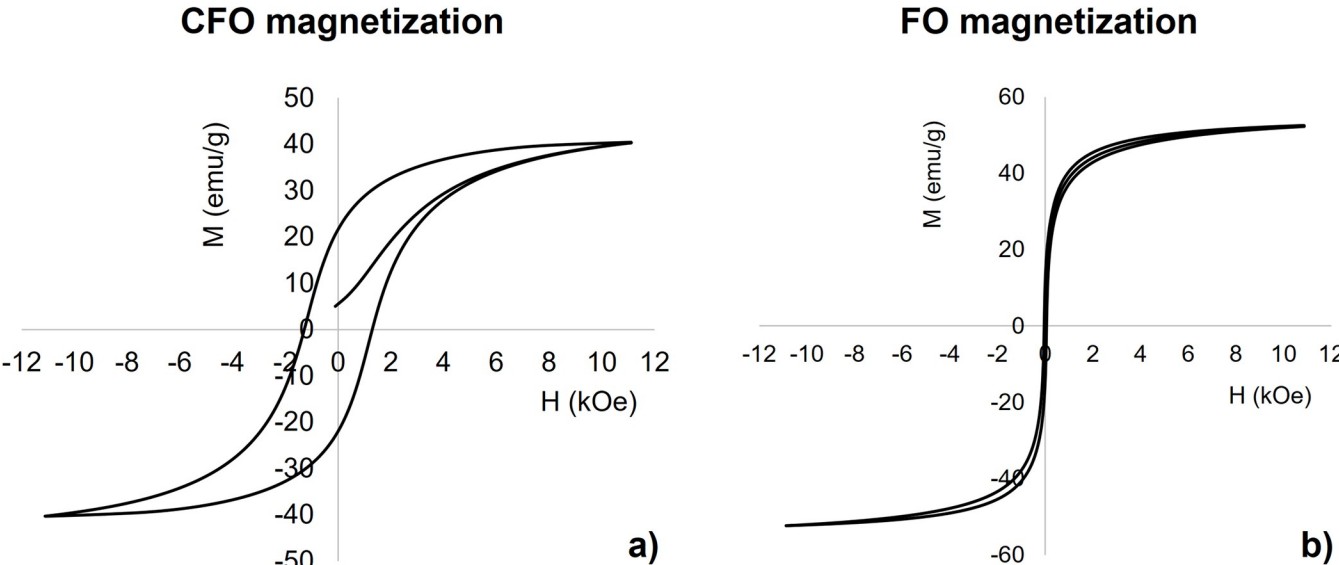

**Fig 4. Effect of core and shell size on magnetoelectric coefficient.** Trend analysis of variable core size (a and b) and shell thickness (c and d) of CFO-BTO (a and c) and FO-BTO (b and d) MENPs when stimulated with a high strength (> Ms) DC bias magnetic field directed along z on the magnetoelectric coefficient $\alpha_{ME}$ (V/cm·Oe).

(H(Oe), blue line) directed along z and composed by a high amplitude (H = 10 kOe) 2 seconds DC bias excitation, follwed by a sinusoidal weak magnetic field excitation (f = 50 Hz, 100 Oe) lasting the remaining 4 seconds. When the DC bias is switched off (after 2 seconds), the core magnetization decreases according to the hysteretic behavior shown in Fig 2A. Once the low intensity AC magnetic field is applied, magnetization starts again to increase following the sinusoidal behavior of the excitation signal (see e.g. the magnification of Fig 5A between 3 and 3.1 s shown in Fig 5B). This process initiates the core magnetostriction and the consequent induced strain in the shell that in turn produces a time-variant electric field around the nanoparticle.

According to our results, the electric field generated by a 50 nm CFO-15 nm BTO nanoparticle, can reach values above $10^5$ V/m when DC bias is activated and reaches tens of thousands V/m when DC bias is switched off and only the weak AC field is applied as excitation. This corresponds to magnetoelectric coefficients, calculated considering the maximum change in magnetic field equal to the 100 Oe peak AC amplitude, that can reach up to 15 V/cm·Oe.

## 4. Discussion

Even though the ME effect is known for many decades, only recently materials with strong ME effects at room temperature, which is particularly important in the biomedical field, have been made available.

Among the various mechanism to achieve magnetoelectric coupling, strain-mediated composites, such as core-shell nanoparticles, are one of the most promising for enabling biomedical applications, owing by a large selection of materials, limitless potential geometries, and highly customizable configurability [19,21].

The most popular materials combination of such nanoparticles consists of piezoelectric shell made of barium titanate ($BaTiO_3$, BTO) and magnetostrictive core of cobalt ferrite ($CoFe_2O_4$, CFO), but others iron oxides such as magnetite ($Fe_2O_3$—FO) have been used [50]. To exploit the broad potential applications of these materials a deep knowledge of their

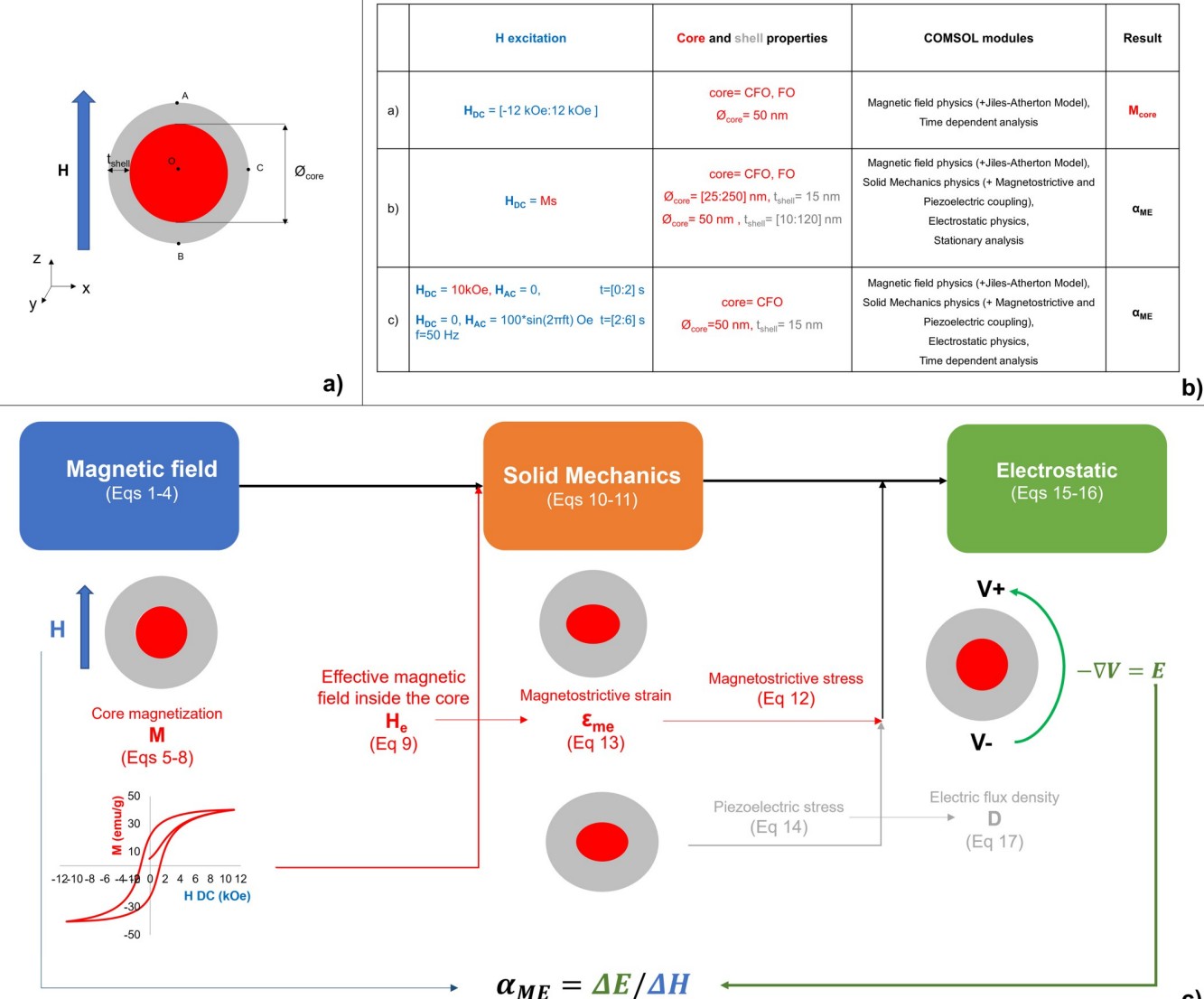

**Fig 5. Magnetization of MENP under DC+AC stimulation.** Magnetization M(emu/g) (red line) of a CFO core ($\varnothing_{core}$ = 50 nm)-BTO shell ($t_{shell}$ = 15 nm) nanoparticle under a DC+AC external magnetic field (H (Oe)- blue line) directed along z. a) M(emu/g) as a function of 2 seconds DC high amplitude (H = 10 kOe) magnetic field followed by 4 seconds weak AC (f = 50 Hz, 100 Oe) magnetic field excitation. b) Magnification of Fig 5A in five AC excitation periods.

characteristics and of the possible ways to control their behavior is mandatory to fit safety requirements and optimize their electric output. In this study we aimed to provide a computational framework that can be applied to assess MENPs behavior for variable stimulation conditions and then for versatile applications, with an eye to their possible use for brain stimulation.

Two core materials (CFO and FO) have been characterized, starting from their magnetic behavior in response to varying DC bias magnetic field. Results of the calculated hysteresis loops (Fig 2) agree with experimental studies [22,36] and confirm that CFO nanoparticles (Fig 2A), despite their small sizes, do not become superparamagnetic at room temperature and that, when the DC bias is removed, remain magnetized. Conversely, FO loop (Fig 2B) suggests a superparamagnetic behavior with no hysteresis and at room temperature those nanoparticles remanent magnetization approaches to zero [37,51]. This different performance is due to the

significant difference, by approximately an order of magnitude, between their magneto crystalline anisotropy energies and corresponding magnetic core stability ratios and translates in magnetostriction coefficients that differ in one order of magnitude as well [52]. In order to optimize this aspect, increasing FO nanoparticles core volume and decreasing temperature could be valuable strategies [53], but both not feasible for their use in *in vivo* applications where small nanoparticles at room temperature are injected into the body. To this purpose, Fig 3 quantifies how these different core magnetostriction properties produce different magnetoelectric effects for reasonable average-sized 80 nm (50 nm core diameter and 15 nm shell thickness) core-shell CFO-BTO and FO-BTO nanoparticles when a strong (above saturation) DC bias magnetic field is applied. During the magnetization process (Fig 3A), the cores change their dimension and become compressed along the applied magnetic field direction (i.e. z-axis) and decompressed along the other two dimensions (Fig 3B). This strain is then transferred to the piezoelectric shell, which converts, according to the calculated piezoelectric coefficient $d_{33}$ = 78 pC/N, the stress imposed by the core to a change in charge distribution and then electric field distribution (Fig 3C) and surface potential (Fig 3D). As clearly shown in Fig 3D, the perfect symmetry of the potential distribution, reveals that each nanoparticle, at a given instant, can be modelled as an electric dipole aligned along the magnetic field lines. The electric potential difference between the two poles (A and B) quantifies the magnetoelectric coefficient as $\alpha_{ME} = (V_{AB}/D)/H$ where D is the nanoparticle diameter. This value is equal to 0.28 V/cm·Oe for an 80 nm CFO-BTO nanoparticle with 60:40 core to shell ratio and about one order of magnitude lower (i.e. 0.032 V/cm Oe) for a FO-BTO nanoparticle of the same dimension and core-shell composition. Both those values agree with magnetoelectric coefficients estimated on nanoparticles of similar sizes [16,28,53,54] and are about one order of magnitude higher than experimental values [20,55]. This discrepancy is due to the not perfect coupling between the two phases at the interface, which is not accounted when using ideal parameters and perfectly matching boundary conditions as in the simulations [56].

As extensively discussed in literature [18,24,25,27,28,53,57], the structure of the boundary surface, including its extension, indeed strongly affects the ME effect in a way such that higher ME coefficient will require large core-shell ratios. Our analysis on variable MENP core and shell size (Fig 4) further reinforces this concept and reveals that is possible to maximize the ME effect by increasing the magnetostrictive phase content and by reducing the shell thickness. Larger core volumes produce indeed larger magnetostrictive strains and the yielded compressive strengths are more efficiently transferred to the outer boundary of the shell when its thickness is reduced.

These theoretical results need however to be framed into the special cases for which specific MENPs are developed. In the context of *in vivo* applications, such as in brain stimulation and drug delivery, one should consider that most of endothelial barriers allow nanoparticles up to 200 nm in diameter to pass, but blood brain barrier (BBB), for instance, is much more restrictive [58], allowing only small nanoparticles ($\approx$30 nm in diameter) to reach the brain. Similar considerations held for applications in which MENPs cell membrane penetration is desired. Stimulation conditions could help to solve these issues. In recent studies [31,59,60], high magnetic field gradients (>1000 Oe/cm) have been applied to navigate the nanoparticles in the circulating blood and pull them into the brain, thus proving that tissue delivery can be enhanced by exploiting the magnetic properties of the MENPs core.

Moreover, results of this study also reveal that the application of low amplitude AC magnetic field, after core pre-magnetization through DC bias, can be leveraged for tuning and optimizing the electric output. Most of biomedical applications in which MENPs can be promisingly used, such as brain stimulation, cell proliferation, drug delivery, are enabled by time-variant electric fields. On the other hand, to not violating safety limits in power deposition

within human tissues, it is preferable to limit both the electric field amplitude and its frequency. The analysis of ME response to weak and near-DC time variant magnetic field (see Fig 5) suggests that the "memory effect" of some materials with hysteresis allows to keep the induced alternating electric fields at levels comparable to the ones induces by strong DC bias magnetic field. These results, not only corroborate experimental results where MENPs stimulated by low amplitude AC frequencies are successfully and safely used [27,29–34], but also provide a basis for understanding the ME effect elicited by low energy external magnetic fields and tuning the corresponding electric output.

## 5. Conclusions

Magnetoelectric nanoparticles could represent a potential game-changing tool in biomedicine, in particular for brain stimulation applications, by proving minimally invasive access to human tissues. In this study, we performed an in-silico characterization of recently developed MENPs for biomedical applications and investigated how their composition, dimension as well as the external magnetic field features (namely amplitude and frequency) affect their magnetoelectric response in terms of magnetoelectric coefficient magnitude. The values here found agree with values measured and calculated in experimental studies on MENPs with same properties and stimulated at the same conditions, thus validating our model. Moreover, our results provide additional information that can be strategically used to adapt their use to specific biomedical applications, where safety issues still hamper their performance. We found that the electric output not only depends on the DC bias magnetic field strength but also on its history and that this property can be exploited when stimulating MENPs with low amplitude AC fields. Moreover, higher ME coefficients, with all other conditions being the same, can be obtained by increasing nanoparticles core volume and reducing shell thickness.

The analytical framework here used to assess the behavior of existing core-shell MENPs, can be applied to other geometries and material configurations thus enabling the validation of future novel applications of these materials.

## Supporting information

**S1 Data.**
(ZIP)

## Author Contributions

**Conceptualization:** Serena Fiocchi, Emma Chiaramello, Paolo Ravazzani.

**Data curation:** Serena Fiocchi, Emma Chiaramello, Alessandra Marrella, Marta Bonato.

**Formal analysis:** Serena Fiocchi, Emma Chiaramello, Alessandra Marrella, Giulia Suarato, Marta Parazzini.

**Investigation:** Serena Fiocchi, Emma Chiaramello, Marta Parazzini, Paolo Ravazzani.

**Methodology:** Serena Fiocchi, Emma Chiaramello, Alessandra Marrella, Giulia Suarato, Marta Bonato, Marta Parazzini, Paolo Ravazzani.

**Software:** Serena Fiocchi, Emma Chiaramello, Alessandra Marrella, Marta Bonato.

**Supervision:** Marta Parazzini, Paolo Ravazzani.

**Visualization:** Serena Fiocchi, Emma Chiaramello.

**Writing – original draft:** Serena Fiocchi, Emma Chiaramello.

**Writing – review & editing:** Serena Fiocchi, Emma Chiaramello, Alessandra Marrella, Giulia Suarato, Marta Bonato, Marta Parazzini, Paolo Ravazzani.

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
