## [Decision Letter · Decision Letter 0]

1 Aug 2022

PONE-D-22-20606Modeling of core-shell magneto-electric nanoparticles for biomedical applications: effect of composition, dimension, and magnetic field features on magnetoelectric responsePLOS ONE

Dear Dr. Fiocchi,

Thank you for submitting your manuscript to PLOS ONE. After careful consideration, we feel that it has merit but does not fully meet PLOS ONE’s publication criteria as it currently stands. Therefore, we invite you to submit a revised version of the manuscript that addresses the points raised during the review process.

We look forward to receiving your revised manuscript.

Kind regards,

A M Mansour, Ph.D.

Academic Editor

PLOS ONE

Journal Requirements:

2. Please note that PLOS ONE has specific guidelines on code sharing for submissions in which author-generated code underpins the findings in the manuscript. In these cases, all author-generated code must be made available without restrictions upon publication of the work. Please review our guidelines at https://journals.plos.org/plosone/s/materials-and-software-sharing#loc-sharing-code and ensure that your code is shared in a way that follows best practice and facilitates reproducibility and reuse. New software must comply with the Open Source Definition.

Reviewers' comments:

Reviewer's Responses to Questions

**Comments to the Author**

1. Is the manuscript technically sound, and do the data support the conclusions?

Reviewer #1: Yes

2. Has the statistical analysis been performed appropriately and rigorously? 

Reviewer #1: Yes

3. Have the authors made all data underlying the findings in their manuscript fully available?

Reviewer #1: Yes

4. Is the manuscript presented in an intelligible fashion and written in standard English?

Reviewer #1: Yes

5. Review Comments to the Author

Reviewer #1: This work has a potential and my comments are as follows:

- First of all, the Abstract should contain answers to the following question: why is this investigation important?

- The Introduction should make a compelling case for why the magnetoelectric nanoparticles is beneficial with a clear statement of its novelty and originality by providing relevant information and providing answers in connection with nanoelectrodes.

- What are the special cases of your study?

- Article needs proofreading to eliminate minor typos.

- Basic equations need to be referenced.

- Conclusions should contain an emphasis on the comparison between this work and others for validation.

- Punctuation is missing after some equations.

- For enhancing the introduction section with the new publications, old references may be replaced with new ones such as:

Dynamism of a hybrid Casson nanofluid with laser radiation and chemical reaction through sinusoidal channels

On the entropy optimization of hemodynamic peristaltic pumping of a nanofluid with geometry effects

Performance enhancement of a DC-operated micropump with electroosmosis in a hybrid nanofluid: fractional Cattaneo heat flux problem

Three-dimensional nanofluid stirring with non-uniform heat source/sink through an elongated sheet

Computational Framework of Magnetized MgO–Ni/Water-Based Stagnation Nanoflow Past an Elastic Stretching Surface: Application in Solar Energy Coatings

Biomedical simulations of nanoparticles drug delivery to blood hemodynamics in diseased organs: Synovitis problem

Leveraging elasticity to uncover the role of rabinowitsch suspension through a wavelike conduit: Consolidated blood suspension application

6. PLOS authors have the option to publish the peer review history of their article (what does this mean?). If published, this will include your full peer review and any attached files.

Reviewer #1: No

---

## [Author Response · Author response to Decision Letter 0]

26 Aug 2022

We have thoroughly revised the manuscript, addressed all queries one by one and made corrections throughout the text and corresponding figures/tables. All the changes of the original manuscript have been reported in the response to the decision letter attached (Response to reviewers.docx) and have been highlighted in red in the revised manuscript.

---

## [Editor Report · Decision Letter 1]

2 Sep 2022

Modeling of core-shell magneto-electric nanoparticles for biomedical applications: effect of composition, dimension, and magnetic field features on magnetoelectric response

PONE-D-22-20606R1

Dear Dr. Fiocchi,

We’re pleased to inform you that your manuscript has been judged scientifically suitable for publication and will be formally accepted for publication once it meets all outstanding technical requirements.

Kind regards,

A M Mansour, Ph.D.

Academic Editor

PLOS ONE
---

## [Editor Report · Acceptance letter]

14 Sep 2022

PONE-D-22-20606R1 

Modeling of core-shell magneto-electric nanoparticles for biomedical applications: effect of composition, dimension, and magnetic field features on magnetoelectric response 

Dear Dr. Fiocchi:

I'm pleased to inform you that your manuscript has been deemed suitable for publication in PLOS ONE. Congratulations! Your manuscript is now with our production department. 

Kind regards, 

on behalf of

Prof A M Mansour 

Academic Editor

PLOS ONE